# Complement Regulation in Human Tenocytes under the Influence of Anaphylatoxin C5a

**DOI:** 10.3390/ijms22063105

**Published:** 2021-03-18

**Authors:** Sandeep Silawal, Benjamin Kohl, Jingjian Shi, Gundula Schulze-Tanzil

**Affiliations:** 1Institute of Anatomy and Cell Biology, Paracelsus Medical University, Nuremberg and Salzburg, General Hospital Nuremberg, Prof. Ernst Nathan Str. 1, 90419 Nuremberg, Germany; sandeep.silawal@pmu.ac.at (S.S.); shijingjian55@126.com (J.S.); 2Department of Traumatology and Reconstructive Surgery, Campus Benjamin Franklin, Charité—Universitätsmedizin Berlin, Corporate Member of Berlin Institute of Health, Freie Universität Berlin and Humboldt-Universität zu Berlin, Hindenburgdamm 30, 12203 Berlin, Germany; benjamin.kohl@charite.de

**Keywords:** anaphylatoxin, C5a, complement system, complement regulation, tendinopathy, tenocytes

## Abstract

A central part of the complement system, the anaphylatoxin C5a was investigated in this study to learn its effects on tenocytes in respect to understanding the potential expression of other crucial complement factors and pro-inflammatory mediators involved in tendinopathy. Human hamstring tendon-derived tenocytes were treated with recombinant C5a protein in concentrations of 25 ng/mL and 100 ng/mL for 0.5 h (early phase), 4 h (intermediate phase), and 24 h (late phase). Tenocytes survival was assessed after 24 h stimulation by live-dead assay. The gene expression of complement-related factors C5aR, the complement regulatory proteins (CRPs) CD46, CD55, CD59, and of the pro-inflammatory cytokines tumor necrosis factor (TNF)-α and interleukin (IL)-6 was monitored using qPCR. Tenocytes were immunolabeled for C5aR and CD55 proteins. TNFα production was monitored by ELISA. Tenocyte survival was not impaired through C5a stimulation. Interestingly, the gene expression of C5aR and that of the CRPs CD46 and CD59 was significantly reduced in the intermediate and late phase, and that of TNFα only in an early phase, compared to the control group. ELISA analysis indicated a concomitant not significant trend of impaired TNFα protein synthesis at 4 h. However, there was also an early significant induction of CD55 and CD59 mediated by 25 ng/mL anaphylatoxin C5a. Hence, exposure of tenocytes to C5a obviously evokes a time and concentration-dependent response in their expression of complement and pro-inflammatory factors. C5a, released in damaged tendons, might directly contribute to tenocyte activation and thereby be involved in tendon healing and tendinopathy.

## 1. Introduction

Tendinopathy has been defined as a chronic low grade inflammatory and degenerative musculoskeletal disease [1,2]. Even though the etiology of this disease is still not clearly understood, there are various hypotheses that are frequently discussed. The collagen tears resulting from so-called micro traumata after repetitive mechanical overloading have been a long known classical explanation for the pain and functional impairment in tendinopathy [3]. Moreover, the inflammatory concept in the pathogenesis of tendinopathy has regained a wider focus [2]. A low increase in pro-inflammatory cytokines such as tumor necrosis factor (TNF)-α is associated with the pathological tendons and the affected tenocytes [4]. Tenocytes’ activity alters in response to any physiological and non-physiological mechanical loading, biochemical imbalance in their environment, or inflammation, leading to morphological changes and changes in their extracellular matrix synthesis [5,6,7]. In addition, an increased presence of inflammatory cells such as macrophages, mast cells, and T-cells in pathological tendons was emphasized in comparison to healthy tendons in a systematic review [8]. These immune cells are actively involved in producing pro-inflammatory cytokines, such as TNFα, interleukin (IL)-6, IL-1β, etc., in addition to complement factors that can trigger the inflammation process [1,9]. A direct TNFα stimulation of tenocytes and an indirect co-culture with leukocyte subpopulations, peripheral blood mononuclear cells, and neutrophils have been studied in our previous in vitro study to understand the complement response in tenocytes [10].

The complement system is an important first line of defense in our immune system. With more than 30 different proteins belonging to this system, its activation can be initiated through the classical, lectin, and spontaneous hydrolysis pathways [11]. The complement protein activation by cleavage leads to commonly known effects such as opsonization, inflammation, and cell lysis (Figure 1).

A common trunk protein complement factor C5 is fragmented through C5 convertase and other extrinsic proteases such as cathepsin D into C5a, a small (9 kDa) but potent so-called anaphylatoxin and C5b. C5b attaches to the cell membrane. The C5b split fragment associates with other complement factors to generate a membrane attack complex (MAC, C5b–C9), leading to either cellular osmotic lysis or other sublytic responses [12,13]. C5a binds to the G-protein coupled membrane protein C5a receptor (C5aR) or C5a receptor-like 2 (C5L2), a non-signaling C5a binding protein, unfolding its effects on the target cells. Hereby, C5a triggers signal cascades in an affected cell after it attaches to C5aR [11]. Inflammatory effects in various tissues have been described as a result of the C5a/C5aR coupling such as their stimulatory effect on leukocytes, including chemotaxis of leukocytes, smooth muscle contraction, or increase in vascular permeability; hence, C5a is known to be a key player in the development of inflammation and sepsis [14,15]. A physiological plasma concentration of 8.34 ± 2.05 ng/mL C5a has been reported [16]. Interestingly, C5a and C3a have also been associated with tissue regeneration and repair [17]. There are various mechanisms regulating complement effects for the protection of our own body cells. These are exerted by so-called complement regulatory proteins (CRPs), which are available in the system either as soluble factors such as C1-inhibitor, factor H, factor H-like protein 1, clusterin, vitronectin, or cell membrane-attached proteins, such as CD46 (membrane co-factor), CD55 (decay-accelerating factor), CD59 (protectin), etc. [11]. Complement regulatory proteins CD46 and CD55 regulate the complement cascade in the early cascade level at which they attenuate the C3 convertase or accelerate its degradation preventing the complement activation in the intermediate stage. CD59, on the other hand, hinders the association of membrane attack complex (MAC) as the terminal component of the complement cascade.

TNFα and other pro-inflammatory mediators often accompany an increase in complement and can regulate each other in an inflammatory process [10,18,19]. Even though various cytokines have been discussed in the research field of tendon inflammation [20,21], IL-1β, TNFα, IL-6, and IL-10 are considered in various studies as the key cytokines in tendon diseases, as reported in a systematic review [9]. The complement system attracts increasing attention in this tendon research since complement factors and receptors could be detected in tenocytes and their regulation in-vitro in response to physical and biochemical influences [10,18,19]. Therefore, more research has to be performed to understand the role of complement factors as possible players in the pathogenesis of tendinopathy.

## 2. Results

### 2.1. Tenocyte Survive C5a Exposition

To assess whether C5a stimulation of tenocytes for 24 h (Figure 2A) impairs cell survival, a vitality assay has been performed visualizing vital cells with green fluorescence and dead cells with red fluorescence (Figure 2B). While areas covered with vital cells compared to those areas covered in total by cells were measured using ImageJ, no significant impairment in tenocytes vitality in response to both concentrations of the anaphylatoxin C5a was seen compared to the control group (Figure 2C).

### 2.2. Tenocytes Show a Time and Concentration Dependent Regulation of C5aR, CRPs, TNFα, and IL-6 Gene Expression

The C5aR gene expression was significantly reduced under both C5a concentrations at the intermediate phase time point of analysis (4 h). This effect could still be detected in the tenocytes at the late phase of 24 h but only, when exposed to the higher concentration of 100 ng/mL C5a (Figure 3A). Likewise, the transcriptional activity of the gene encoding for complement regulatory protein CD46 also showed a significant reduction at the investigation time point of 24 h, but only in response to the high concentration of 100 ng/mL C5a (Figure 3B). In the early phase of C5a stimulation (0.5 h), a significant increase in gene expression of complement regulatory proteins CD55 and CD59 in response to the concentration of 25 ng/mL C5a became evident (Figure 3C,D), but this effect was no longer detectable at later investigation time points.

Interestingly, the gene expression of CD59 decreased significantly at both later investigation time points (4 h and 24 h), irrespective of the C5a concentrations (25 ng/mL or 100 ng/mL) applied (Figure 3D).

The gene expression of the pro-inflammatory cytokine TNFα was significantly suppressed by 100 ng/mL C5a at the 0.5 h stimulation time (Figure 4A). No further significant effect on TNFα gene expression was observed in later phases. IL-6 gene expression was not significantly affected by C5a stimulation (Figure 4B). 

### 2.3. Intracellular TNFα Production in Response to C5a

An intracellular concentration of 4.19 ± 2.78 pg TNFα protein per 1 µg total tenocyte protein was measured in the non-stimulated group at 0.5 h. No significant differences in the TNFα protein concentrations were detected (Figure 5). However, a trend of protein suppression could be observed in the early and intermediate phases of the stimulation. The data in the 24 h stimulation group showed no normal distribution; hence, the interpretation might be speculative.

### 2.4. Immunolabeling of the Proteins C5aR and CD55

The protein expression levels of C5aR and CD55 were analyzed using immunostaining images of these proteins. Despite not reaching the level of significance, we could only observe a trend of a decrease in both C5aR and CD55 protein expression in response to C5a stimulation compared to the respective control groups (Figure 6 and Figure 7). C5aR protein expression of 4 h stimulation with 100 ng/mL C5a is not normally distributed, and hence not qualified for the interpretation. Additional F-actin staining demonstrated the shape and morphology of the tenocytes to better visualize the distribution of the target protein C5aR (Figure 6 and Figure 7). Tenocytes contained F-actin filaments as so-called stress fibers leading to the focal adhesion sites of the cells with no detectable difference evoked by the treatment with C5a.

## 3. Discussion

During the complement activation, the complement factors C3, C4, and C5 are split to generate the anaphylatoxins C3a, C4a, and C5a, respectively [11]. The aim of the study is to understand the effect of C5a, known as an inflammatory mediator, on human tenocytes with a particular focus on complement regulation and cytokines TNFα and IL-6. The results should help to hypothesize the putative role of C5a in tendinopathy. The C5 split fragment, C5a, is known to induce various inflammatory cell responses and has also been described as inducing apoptosis in adrenomedullary cells in an animal model of sepsis [15,22]. Additionally, a high concentration of 50 ng/mL recombinant C5a treatment of mouse kidney endothelial cells showed significantly higher rates of apoptosis than the lower concentrations or the control [23]. However, the vitality of human tenocytes exposed to concentrations of 25 ng/mL and 100 ng/mL C5a did not show significant changes; hence, cell vitality remained mainly unaffected. The selected concentration of 25 ng/mL C5a is between that used in the above-mentioned study [23] and the C5a amount measured in human plasma (8.34 ± 2.05 ng/mL) [16]. This displays the rather robust nature of the human tenocytes when exposed to these concentrations of anaphylatoxin C5a for the whole treatment time period. However, the tenocytes used in this study were derived from healthy tendons of young donors and one could hypothesize that cells from tendinopathic tendons might display a more sensitive response. The focus of our study is to understand the effect of C5a on tenocytes in regard to the inflammatory processes probably contributing to tendinopathy. The tenocytes react significantly in response to anaphylatoxin C5a with suppression of C5aR gene expression in the intermediate and late phases. No significant effects could be obtained in the early phase (Figure 3A). The semi-quantitative evaluation of C5aR protein expression at 4 h displayed the gene expression pattern as a trend at a similar time point. The decrease of C5aR in gene and protein expression could suggest the self-regulation of the inflammatory effect in tenocytes representing a negative feedback loop.

The CD46 gene expression of tenocytes was suppressed in a very late phase under 100 ng/mL C5a concentration (Figure 3B). The suppression could also be observed after C3a stimulation in a previous study even in an earlier stage (i.e., in an intermediate phase) [18]. Under 25 ng/mL C5a stimulation, CD55 and CD59 gene expression of tenocytes was induced in a very early phase (Figure 3C,D). This shows a possible self-protective capacity of the tenocytes in the early phase against the anaphylatoxin C5a. Interestingly, stimulation of tenocytes by another anaphylatoxin C3a more upstream localized in the complement cascade evoked a decrease in the CD55 gene expression in an intermediate phase [18]. Nonetheless, in our study, CD59 gene expression was reduced in both time periods and in response to both concentrations in the intermediate and the late phases (Figure 3D). This displays an early response of tenocytes to C5a expressing the C5aR and CD59 genes, which probably recedes after 4 h to negative feedback.

The pro-inflammatory cytokine TNFα is available at the site of inflammation, and blocking this agent is a clinical approach to treat various inflammatory diseases. It has been proven that tenocytes are also able to produce TNFα, and in response to TNFα, they react with an upregulation of the gene expression of pro-inflammatory (TNFα, IL-1β) and immunoregulatory (IL-6, IL-10) cytokines [20]. In addition, TNFα is known to be increased in the case of tendon pathology [4]. We know that C5aR gene expression is elevated in tenocytes in response to TNFα stimulation in an intermediate stage (4 h) of inflammation [10]. Conversely, in our present study, we could show that C5a suppresses the TNFα gene expression in these cells in an early phase of 0.5 h in comparison to the control group (Figure 4A). This effect on TNFα could be determined in the intermediate phase of the stimulation (Figure 5) at the protein level by ELISA analysis. In leukocytes, C5a stimulates the synthesis and release of pro-inflammatory cytokines such as TNFα, IL-1β, IL-6, etc. after 24 h [24,25]. However, C5a, as shown in a sepsis in vitro study, also decreased NFκB-dependent gene transcription of TNFα and impaired lipopolysaccharide-induced TNFα production in neutrophils [26]. The effect of TNFα suppression in tenocytes is a novel result. The suppression of both TNFα gene expression in the early phase, followed by a trend of impaired TNFα protein synthesis in the intermediate phase, reduced C5aR gene transcription in the intermediate and late phases, and early induction of CD55 and CD59 gene expression, could open the hypothesis that the tenocytes could have an anti-inflammatory self-regulatory mechanism against the inflammatory effect of anaphylatoxin C5a. TNFα alone can also trigger the IL-6 expression in the tenocytes and also a higher expression of IL-6 has been associated with tendinopathy [20,27]. Even though it is in our highest interest to understand the possible effect of anaphylatoxin C5a on the IL-6 synthesis focusing on tenocytes, in our study, no clear effect regarding IL-6 gene expression in tenocytes in response to this anaphylatoxin was detected (Figure 4B). A previous study suggested that the immunoregulatory cytokine IL-6 is induced through TNFα in tenocytes, whereas the present study could not prove any regulation through C5a in the tenocytes [20]. However, one limitation of this study is that the number of independent experiments performed with different cell donors involved varied at the gene and protein expression analysis, which might be responsible for non-significant results at the protein level.

The treatment of human tenocytes derived from non-inflamed hamstring tendons with anaphylatoxin C5a does not necessarily replicate typical micro-environmental conditions of tendinopathy. Interestingly, tenocytes exposed to C5a showed some anti-inflammatory responses, such as the early suppression of TNFα gene expression. Hence, future research of anaphylatoxin effects on tenocytes should include the presence of pro- or anti-inflammatory cytokines and cells from healthy and affected tendons of different disease stages to allow an even better understanding of the inflammatory aspects of tendinopathy.

## 4. Materials and Methods

### 4.1. Isolation of the Tenocytes and Tenocyte Culturing

Tenocytes derived from human hamstring tendons collected from residual tissue after anterior cruciate ligament replacement surgeries were used in this project. The donors were exclusively males between 18 and 53 years old with an average age of 31.1 years old. Their use was approved by the Charité review board (Charité-Universitätsmedizin Berlin, Campus Benjamin Franklin, Berlin, Germany [Ethical approval number EA4-033-08]). Before cutting the tissue into small pieces (1–4 mm), the paratenon of this tissue was carefully removed. The tissue was cultured in T25 culture flasks (Sarstedt, Hildesheim, Germany) with a growth medium at 37 °C and 5% CO_2_. The growth medium for tenocyte culturing consisted of Dulbecco’s MEM/Ham’s F-12 (1:1 mixture) (Merck KGaA, Darmstadt, Germany), 25 mg/mL ascorbic acid, 50 IU/mL streptomycin, 50 IU/mL penicillin, 2.5 μg/mL amphotericin B, and essential amino acids [all: Merck KGaA,]. A total of 10% fetal calf serum (FCS) was added to the growth medium for cell expansion purposes. The tenocytes emigrated from the explants after circa 7–10 days. To detach the cells from the flask for expansion, 0.05% trypsin/1.0 mM ethylenediaminetetraacetic acid (EDTA) (Merck KGaA) was used. Cells at passages (P)4–7 were used for the experiments.

### 4.2. Stimulation of Tenocytes with Anaphylatoxin C5a

Overall, 10,000 tenocytes/cm^2^ were cultivated for 24 h in 10% FCS containing growth medium and subsequently serum-starved for 1 h in starvation medium containing only 1% FCS before the stimulation experiment was performed. Simultaneously the C5a containing stimulation medium was prepared using 25 ng/mL and 100 ng/mL recombinant human C5a protein (Catalog Number: 2037-C5, R&D Systems, Minneapolis, MN, USA) in the starvation medium. Non-stimulated tenocytes were the control group (Figure 2A).

### 4.3. Vitality Test of the Tenocytes

After stimulation of the tenocytes cultivated on the poly-L-lysine (Merck KGaA) coated coverslips, the vitality staining of tenocytes was performed. The tenocytes were incubated in a mixture of phosphate-buffered saline (PBS) with fluorescein diacetate (final concentration = 15 µg/mL) and propidium iodide (final concentration = 1 µg/mL) (Thermo Fisher Scientific, Rockford, IL, USA). The green fluorescence representing the vital cells or red fluorescence indicating dead cells was visualized using a confocal laser scanning microscope (Leica TCS SPEII and DMi8, Wetzlar, Germany). The images of the tenocytes were analyzed using image processing software (ImageJ, U.S. National Institutes of Health, Bethesda, MD, USA). Here, the area covered by living cells was calculated in relation to the total area colonized by cells (100%). Three independent experiments were included with three microscopic fields for each stimulation.

### 4.4. Measurement of TNFα Release by ELISA

Human tenocyte monolayers were washed twice with PBS after completion of the experiment and were incubated with 300 µL lysis buffer (RIPA buffer, Thermo Fisher Scientific) supplemented with protease inhibitor cocktail tablets (Roche, Indianapolis, IN, USA) for 15 min on ice. The cells were additionally, mechanically lysed using cell scrapers. The lysed cells were diluted with PBS attaining a final volume of 1 mL. The cell lysates were centrifuged for 15 min at 14,000× *g*. The acquired supernatants were transferred to another microcentrifuge tube and stored at −80 °C. The remaining cell debris was discarded. Roti^®^-Nanoquant kit (Carl Roth GmbH&Co.KG, Karlsruhe, Germany) was used to determine the protein concentration following the manufacturer’s protocol. Overall, 5 µg of the total protein was used in each sample to perform the ELISA assay.

The Human TNFα Uncoated ELISA kit (Catalog Number 88-7346, Thermo Fisher Scientific) was used for the ELISA measurements. The 96-well plate provided with the kit was first coated with capture antibody and incubated at 4 °C overnight. On the next day, washing steps were performed with PBS containing 0.05 g/L Tween^®^ 20 (Sigma-Aldrich Chemie GmbH, Munich, Germany). All of the following steps in the assay were executed according to the manufacturer’s instructions. Ultimately, 50 µL of 1 M H_3_PO_4_ (Sigma-Aldrich) into the wells terminated the reaction and the absorbance was measured at 450 nm using an ELISA plate reader (Infinite M200 Pro, Tecan, Männedorf, Switzerland).

### 4.5. Gene Expression Analysis

#### 4.5.1. RNA Isolation and cDNA Synthesis

After each stimulation time of 0.5 h, 4 h, and 24 h, cell culture supernatants were removed and the tenocytes were rinsed with PBS (Merck KGaA). After washing, the cells were lysed RNeasy lysis buffer (Qiagen GmbH, Hilden, Germany) containing one-hundredth of β-mercaptoethanol (Sigma-Aldrich Chemie GmbH). RNeasy mini kit (Qiagen GmbH) was used to extract the total RNA using the manufacturer’s instructions. RNA quantity and purity were subsequently analyzed using RNA 6000 Nano assay (Agilent Technologies, Santa Clara, CA, USA). cDNA synthesis was performed with the QuantiTect Reverse Transcription Kit (Qiagen GmbH) in a Mastercycler (Eppendorf AG, Hamburg, Germany).

#### 4.5.2. qPCR

Real-time detection polymerase chain reaction (qPCR) analyses were performed in 20 µL reaction volume applying the derived cDNA. Specific primers for genes encoding complement components C5aR, CD46, CD55, CD59, and pro-inflammatory cytokines TNFα and IL-6 (Table 1, Applied Biosystems (ABI), Foster City, USA) were used as genes of interest and the reference gene was hypoxanthine phosphoribosyl transferase (HPRT)-1 (Table 1). HPRT1 served as the most stable reference gene tested before by comparing it with other reference genes including β-actin, TATA-binding protein, or glyceraldehyde 3-phosphate dehydrogenase. Together with TaqMan Gene Expression Master Mix solution (Thermo Fisher Scientific, Carlsbad, CA, USA), RNAse free water, respective primer, and 20 ng pro well aliquot of the derived cDNA were mixed to perform the TaqMan Gene Expression Assay. The assays were carried out in Real-Time-Cyclers (Opticon I, Bio-Rad Laboratories Inc., Hercules, CA, USA and Chromo4, Bio-Rad Laboratories, Hercules, CA, USA). As per the manufacturer´s instructions, the TaqMan analyses protocols describe the process as follows: the reaction mixture was heated at 50 °C for 2 min, the temperature rose to 95 °C for 10 min followed by the second stage. This stage runs in 40 cycles with 95 °C for 15 s to denaturate the DNA template and an annealing/ extending step at 60 °C for 1 min. Relative gene expression levels were normalized versus the reference gene. Finally, the calculation was performed using the 2-deltaCT method [28].

### 4.6. Immunofluorescence Labeling of the Proteins C5aR and CD55

After 4 h and 24 h stimulation of the human tenocytes as described in Section 2.2, the cells were fixated in 4% paraformaldehyde (PFA, Morphisto, Frankfurt am Main, Germany). After rinsing the cells in Tris-buffered saline (TBS) they were incubated in blocking buffer solution (5% protease-free donkey serum diluted in TBS with 0.1% Triton ×100) at room temperature (RT) for 20 min. The coverslips were rinsed thoroughly and incubated with the primary antibodies directed against C5aR1 and CD55 (for detailed information of antibodies and dyes, see Table 2) in a humidifier chamber overnight at 4 °C. Once again, coverslips were washed thoroughly with TBS before incubation with donkey anti-mouse Alexa Fluor 488 and donkey-anti-goat Cyanine (Cy)3 (Table 2), coupled with secondary antibodies for 60 min at RT. Cell nuclei were counterstained using 4′,6-diamidino-2-phenylindole (DAPI) and the cytoskeleton with Phalloidin Alexa Fluor 633 (Table 2). Negative controls were performed omitting the primary antibody during the staining procedure. Isotype controls were also performed using goat IgG isotype antibody and mouse IgG1 isotype antibody (Table 2) as primary antibodies. Coverslips were washed three times with TBS before being covered with Fluoromount G (Southern Biotech, Birmingham, AL, USA). The images of cells with immunolabeled proteins (three microscopical fields) were subsequently prepared using confocal laser scanning microscopy (Leica). The cell fluorescence intensity of each immunolabeled protein was then measured by an image processing software (ImageJ) [29]. For a comparative analysis of cellular protein synthesis, corrected total cell fluorescence (CTCF) values were measured through which the overlapping background readings over the total cell covered area are deduced from the measured integrated fluorescence density [30,31]. CTCF = integrated fluorescence density—(area of selected cells X mean fluorescence of background readings).

### 4.7. Statistical Analysis

GraphPad Prism (Version 8.1.4), GraphPad Software, San Diego, CA, USA was used for the statistical analysis. The normalized data were expressed as the mean and the standard error of mean (mean ± SEM). Shapiro Wilk normality test and/or Kolmogorov–Smirnov test normality tests were performed. The differences between experimental groups were considered significant at *p* < 0.05 as determined by one sampled *t*-test. * = *p* ≤ 0.05; ** = *p* ≤ 0.01. The Rout test was performed to identify the outliers. The outliers were excluded from the statistics.

## 5. Conclusions

The presence of anaphylatoxin C5a can influence the expression of complement proteins and TNFα in tenocytes. Gene expression of the respective anaphylatoxin receptor C5aR was significantly reduced in 4 h (intermediate) and 24 h (late phase) and that of TNFα at 0.5 h (early phase). Moreover, early induction of CD55 and CD59 gene expression was found in response to the 25 ng/mL concentration of the anaphylatoxin C5a. All these data suggest an anti-inflammatory self-regulatory mechanism of the tenocytes against the inflammatory effect of anaphylatoxin C5a. However, lowered CRPs CD46 and CD59 gene expression in intermediate and late phases imply rather an inflammatory response by the tenocytes; hence, revealing a time and concentration-dependent reaction of the tenocytes to C5a. This study performed under non-inflammatory pre-conditions and void of additional external stressors allows for an understanding of the complement regulation and TNFα-related response of tenocytes directly exposed to this anaphylatoxin.

## Figures and Tables

**Figure 1 ijms-22-03105-f001:**
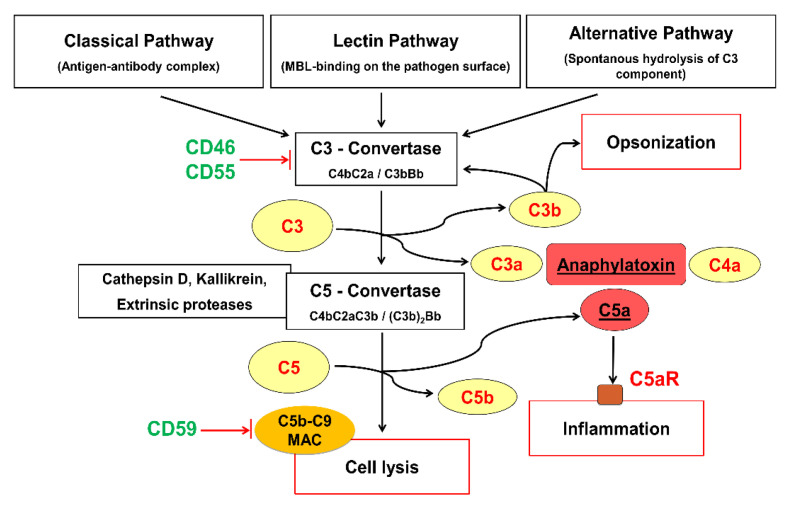
Simplified scheme of the complement activation and regulation processes. C5aR: C5a receptor, MAC: Membrane attack complex, MBL: mannose binding lectin. The image was self-made by Sandeep Silawal.

**Figure 2 ijms-22-03105-f002:**
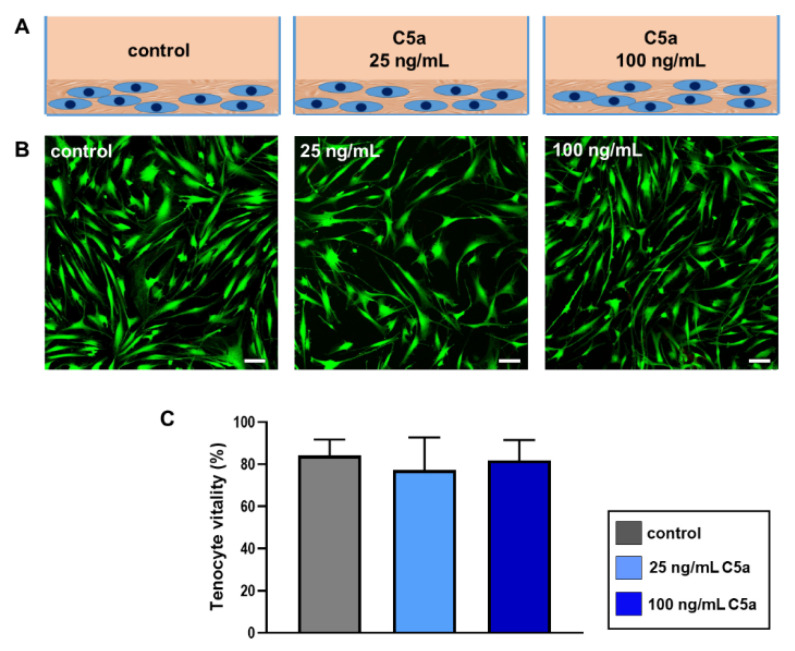
Schematic figure of the experimental setup (**A**). Tenocytes were stimulated with two concentrations (25 ng/mL and 100 ng/mL) of the recombinant protein anaphylatoxin C5a or remained unstimulated (control) for 24 h. (**B**) Representative images of live-dead staining of the tenocytes after 24 h stimulation with recombinant protein anaphylatoxin C5a. Fluorescein diacetate (green: vital cells) and propidium iodide (red: dead cells) have been used for staining. Scale bar = 100 µm. (**C**) Graphic presentation of tenocytes vitality after 24 h stimulation with 25 ng/mL (light blue) and 100 ng/mL (dark blue) recombinant C5a protein in comparison to the non-stimulated control groups (grey). Number of living cells in relation to total cells (100%) measured in live-dead staining images of the tenocytes. Mean values and the standard errors of mean (SEM) are depicted. Shapiro–Wilk test was used to test the normal distribution of data. One sample *t*-test was performed in relation to the theoretical mean of 100%. *n* = 3 independent experiments were performed using tenocytes derived from three different donors.

**Figure 3 ijms-22-03105-f003:**
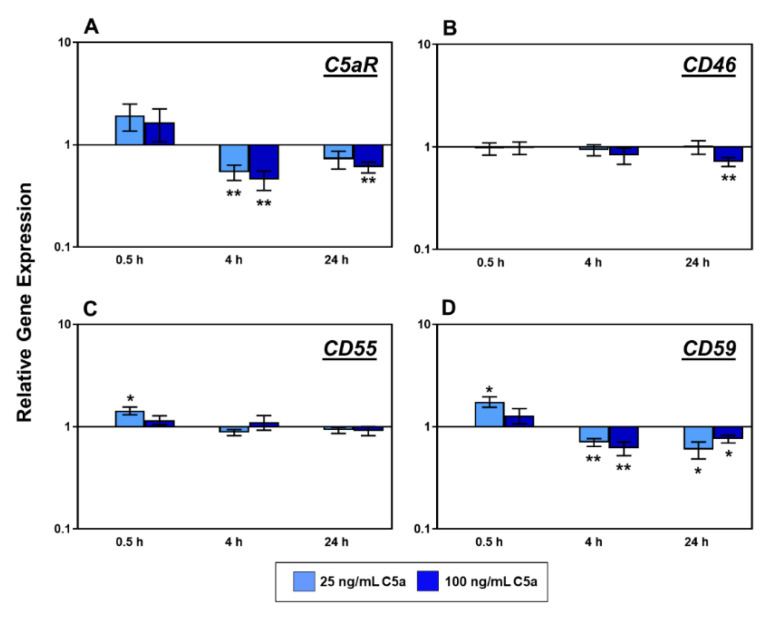
Tenocyte gene expression of *C5aR* (**A**), *CD46* (**B**), *CD55* (**C**) and *CD59* (**D**) after 0.5 h (*n* = 8), 4 h (*n* = 7), and 24 h (*n* = 6) stimulation with 25 ng/mL (light blue) and 100 ng/mL (dark blue) recombinant C5a protein in comparison to the non-stimulated control groups (normalized to 1, horizontal line). Mean values and the standard errors of mean are shown. Gene expression plotted as relative changes with logarithmic scale of Y-axes. Rout outlier test Q = 1.0%. Shapiro Wilk and Kolmogorov–Smirnov normality tests were performed to prove the normal distribution of data. One sample *t*-test with significance in relation to control (*). * = *p* ≤ 0.05; ** = *p* ≤ 0.01. 6–8 independent experiments with human tenocytes derived from different donors were performed.

**Figure 4 ijms-22-03105-f004:**
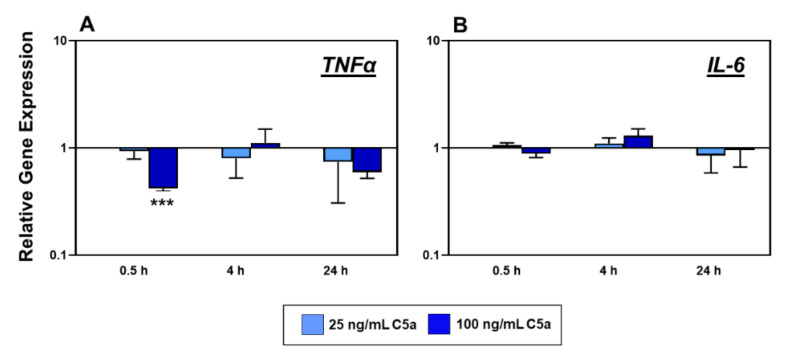
Tenocyte gene expression of *TNFα* (**A**) and *IL-6* (**B**) after 0.5 h, 4 h, and 24 h stimulation with 25 ng/mL (light blue) and 100 ng/mL (dark blue) recombinant C5a protein in comparison to the non-stimulated control groups (normalized to 1, horizontal line). Mean values and the standard errors of mean are shown. Controls have been normalized to 1. Gene expression plotted as relative changes with logarithmic scale of Y-axes. Rout outlier test Q = 1.0%. Shapiro Wilk and Kolmogorov–Smirnov normality tests were performed to prove the normal distribution of data. One sample *t*-test with significance in relation to control (*). *** = *p* ≤ 0.001. *n* = 5 independent experiments with human tenocytes from different donors were performed.

**Figure 5 ijms-22-03105-f005:**
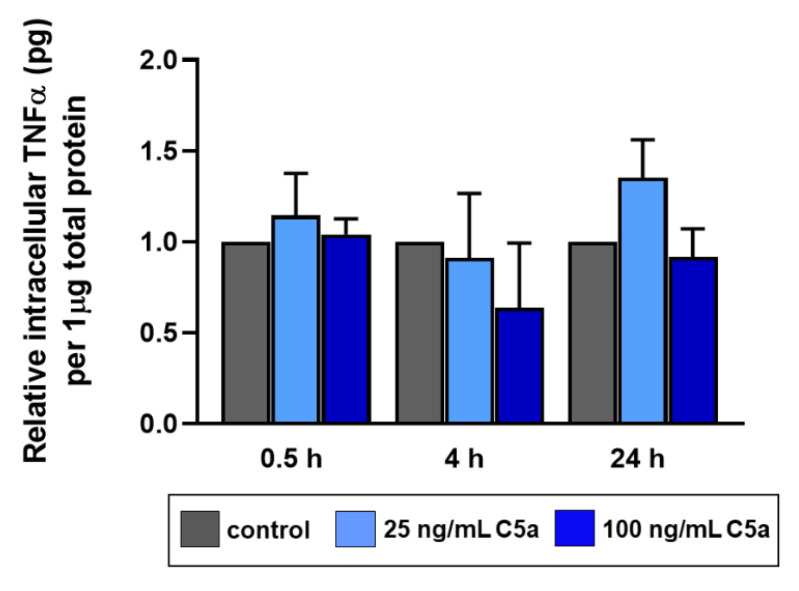
ELISA analysis of intracellular TNFα after 0.5 h, 4 h, and 24 h stimulation with 25 ng/mL (light blue) and 100 ng/mL (dark blue) recombinant C5a protein in comparison to the control group (grey). The values were measured in pg per 1 µg of cell lysate total protein. Mean values and the standard errors of mean are shown. Controls have been normalized to 1. Rout outlier test Q = 1.0%. Shapiro Wilk normality test was used to evaluate the normal distribution of data. One sample *t*-test was performed. *n* = 3 independent experiments with human tenocytes from different donors were performed.

**Figure 6 ijms-22-03105-f006:**
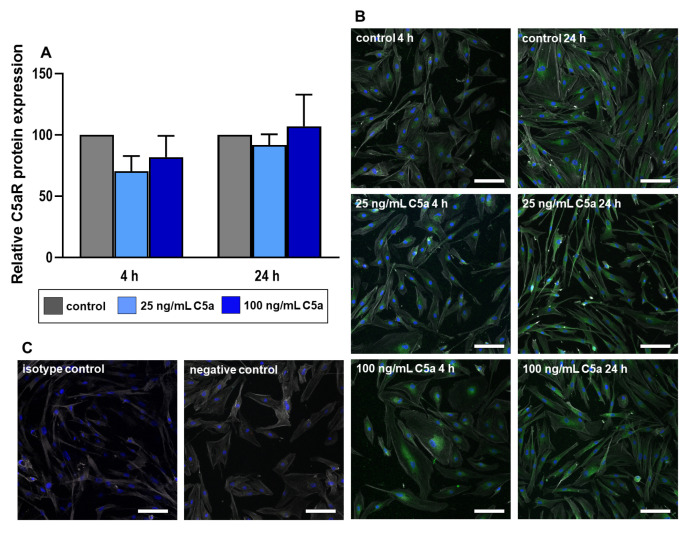
Tenocyte protein expression of C5aR after 4 and 24 h stimulation with 25 ng/mL (light blue) and 100 ng/mL (dark blue) recombinant C5a protein in comparison to the control group (grey) (**A**). Fluorescence intensity indicating C5aR immunoreactivity was analyzed using ImageJ. Mean values and the standard errors of mean are shown. Controls have been normalized to 1. Rout outlier test Q = 1.0%. Shapiro Wilk normality test was used to test the normal distribution of data. One sample *t*-test was performed in relation to control. (**B**) Representative images of tenocytes immunolabeled with C5aR specific antibodies. Green (Alexa 488) = C5aR, blue (DAPI) = cell nuclei, grey (Phalloidin Alexa 633) = actin cytoskeleton. Scale bar = 100 µm. (**C**) Isotype & negative controls of the immunofluorescence staining. *n* = 4 independent experiments with human tenocytes from different donors were performed.

**Figure 7 ijms-22-03105-f007:**
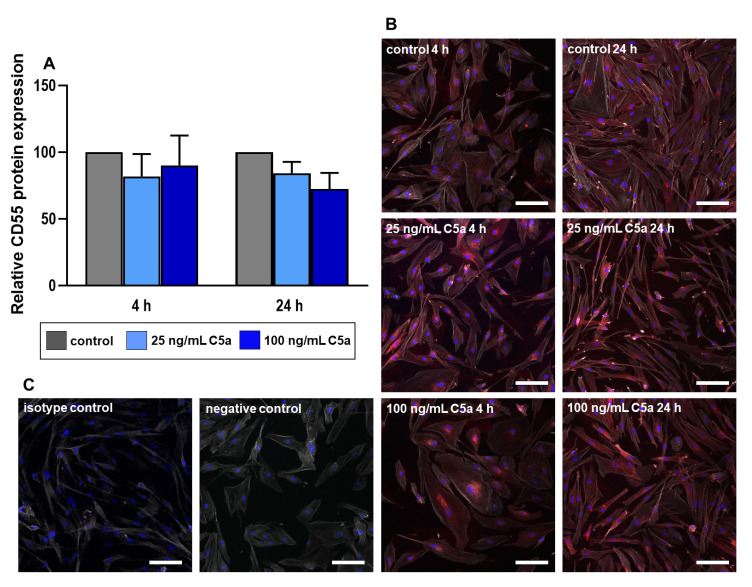
Tenocyte protein expression of CD55 after 4 and 24 h stimulation with 25 ng/mL (light blue) and 100 ng/mL (dark blue) recombinant C5a protein in comparison to the control group (grey) (**A**). Fluorescence intensity indicating CD55 immunoreactivity was analyzed using ImageJ. Mean values and the standard errors of mean are shown. Controls have been normalized to 1. Rout outlier test Q = 1.0%. Shapiro Wilk normality test was used to test the normal distribution of data. One sample *t*-test was performed in relation to control. (**B**) Representative images of tenocytes immunolabeled with CD55 specific antibodies. Red (Cy3) = CD55, blue (DAPI) = cell nuclei, grey (Phalloidin Alexa 633) = actin cytoskeleton. Scale bar = 100 µm. (**C**) Isotype & negative controls of the immunofluorescence staining. *n* = 4 independent experiments with tenocytes from different donors.

**Table 1 ijms-22-03105-t001:** Oligonucleotides used for qPCR analysis.

Primer	Company	Sequence	Probe/Reference	bp
HPRT1	ABI	*	NM_000194.2	100
C5aR	ABI	*	NM_001736.3	68
CD46	ABI	*	NM_172351.1	94
CD55	ABI	*	NM_000574.2	62
CD59	ABI	*	NM_203331.1	70
TNFα	ABI	*	NM_000594.3	80
IL6	ABI	*	NM_000600.4	95

*: Sequence not provided by the company (ABI).

**Table 2 ijms-22-03105-t002:** Antibodies and dyes used.

Specificity and Species	Company	Cat. No.	Stock Concentration	Used Dilution
Goat anti-human CD55 (DAF)	R&D systems, Minneapolis, MN, USA	AF2009	200 µg/mL	1:20
Mouse anti-human C5aR1 (CD88)	GeneTex, Biozol, Eching, Germany	GTX74845	1 mg/mL	1:20
Donkey anti-mouse-Alexa Fluor 488	Thermo Fisher Scientific, Rockford, IL, USA	A21202	2 mg/mL	1:200
Donkey anti-goat-Cy3	Jackson Immuno Research, Cambridgeshire, UK	705165147	1.5 mg/mL	1:200
Goat IgG isotype	Thermo Fisher Scientific, Rockford, IL, USA	02-6202	5 mg/mL	1:100
Mouse IgG1 isotype	Thermo Fisher Scientific, Rockford, IL, USA	02-6102	5 mg/mL	1:100
Phalloidin Alexa Fluor 633	Thermo Fisher Scientific, Rockford, IL, USA	A22284	300 µg/mL	1:1000
4′,6-diamidino-2-phenylindole (DAPI)	Roche Diagnostics GmbH, Basel, Switzerland	10236276001	1 μg/mL	1:100

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
