# Peer review of "Complement Regulation in Human Tenocytes under the Influence of Anaphylatoxin C5a"

_ijms, 2021, doi:10.3390/ijms22063105_

Round 1
Reviewer 1 Report
Manuscript ID: ijms-1127137
Title: Complement regulation in Human tenocytes under the influence of anaphylatoxin C5a
Reviewer comments
This manuscript focuses on one of the intrinsic mechanisms that trigger inflammation that might lead to apoptosis. The source of pro-inflammatory response is mechanical damage to the tendon, associated with the “wear and tear” of the collagen ligaments known as tendinopathy. The stimulation of the tenocytes with the C5a protein, a central component of the pro-inflammatory activation, does not result in any appreciable response from the cells. It indicates that the inflammation developing in the injured tendons is not triggered by the pathway investigated in this work. This information might be of value to other researchers researching this area.
The manuscript is of high quality, and authors are applauded, especially for the attention to detail and preparation of high-quality illustrations.
Specific comments
Line 56: “through the classical, lectin way and spontaneous hydrolysis way [10].” Lectin pathway. The second “way” after the hydrolysis is redundant.
Author Response
The authors would like to thank the reviewer for carefully reading the text and their constructive valuable comments. We modified the manuscript according to their suggestions and comments. A list of changes is reported below. All changes performed are indicated in red and underlined in the revised version of the manuscript. The manuscript has been carefully proof read. Please refer to our point by point reply below. We hope you will find the manuscript suitable for publication in IJMS now.
Sincerely,
Univ.-Prof. Dr. Gundula Schulze-Tanzil
(corresponding author)
This manuscript focuses on one of the intrinsic mechanisms that trigger inflammation that might lead to apoptosis. The source of pro-inflammatory response is mechanical damage to the tendon, associated with the “wear and tear” of the collagen ligaments known as tendinopathy. The stimulation of the tenocytes with the C5a protein, a central component of the pro-inflammatory activation, does not result in any appreciable response from the cells. It indicates that the inflammation developing in the injured tendons is not triggered by the pathway investigated in this work. This information might be of value to other researchers researching this area.
The manuscript is of high quality, and authors are applauded, especially for the attention to detail and preparation of high-quality illustrations.
Specific comments
Line 56: “through the classical, lectin way and spontaneous hydrolysis way [10].” Lectin pathway. The second “way” after the hydrolysis is redundant.
Response: Line 56: Thanks to the reviewer. The redundant word has been eliminated and the sentence has been rewritten as: “… through the classical, lectin and spontaneous hydrolysis pathway [10].”

Reviewer 2 Report
The manuscript by Silawal and colleagues deals with an interesting and relevant topic in the field of tendon research. The results and conclusions drawn are partially based on non-significant data, which should be clearly attenuated overall in the text. I believe that a higher n-value for example in the protein data would lead to similar significant differences comparable to the gene expression data and I would recommend to repeat experiments to strengthen the conclusions. Parts of the text are unclear and I would recommend a professional proofreading. In the text a further publication by Silawal et al. is mentioned which is currently in press and is dealing with the same topic. Therefore, I would like to know the difference to the submitted manuscript. Further detailed comments are found below.
Decision: major revision
Abstract:
The abstract is clearly written and highlights the most important results of the study.
- Line 24: I consider revising the sentence "Similarly, CD46 and …" as the word order is rather confusing.
- Line 28: It says that "C5a, released from damaged tendons". Is this a result of the study, a hypothesis or is it known from literature? If it is known from literature, it is an important fact that should be mentioned at the beginning of the abstract as well as in the introduction.
Introduction:
- The introduction could be in some parts tightened up, as there are doublings in the text that could be shortened (e.g. line 36-38 vs. line 45-48).
Results:
- 2.1.
- Figure 2A: The schematic figure could be improved by drawing not only one cell per well, but some more.
- Figure 2B: the scale bar is missing in the control and 25ng/ml image
- Figure 2C: The very small drop in cell vitality with highest standard deviation is described as "slight decrease". I assume this is a small biological variation rather than a biological effect. Therefore, this should not be mentioned.
- Was the vitality test also performed for the 0.5 h and 4h time point and what was the result? And if not, why?
- 2.2:
- The order of the text should be revised, so that at first figure 3A and B are described and later on C and D.
- Line 134: Figure 2 must be figure 3 and all following figures until figure 6 have the wrong numbering.
- Figure 3: Description of the IL6 gene expression results are missing in the text.
- Regarding all gene expression graphs the y-axis should be displayed in a logarithmic scale to give down-regulations the same distance from 1 as up-regulations.
- 2.3.
- Line 155-156: Mentioning the average protein values above all groups is confusing. What is this good for?
- Describing changes on protein level in % is meaningless if the y-axis is given as fold changes. Changes should be described either as x-fold increase or the y-axis changed to %.
- No significant differences occurred on protein level (probably due to the low number of repetitions n=3) but the description is very detailed. The changes should not be over-interpreted and it should be mentioned that the result are not statistically significant. Compared to e.g. the TNFa gene expression were it is only described that "no further significant effect was observed" every little regulation is described on protein level. Why? If the authors believe that this has a significant biological relevance, the protein experiments should be repeated to increase the n-value and with this the power of the results.
- 2.4.:
- Again, the authors should avoid over-interpreting their data without significance. Please attenuate the statements.
Discussion:
- The discussion would profit from a small paragraph at the beginning introducing again the aim and hypothesis of the study.
- In other cells C5a induced apoptosis. Why was his not reached in your study? Was the concentration different? How were the concentrations defined? Could higher concentrations have led to a reduced cell vitality?
- Line 215-219: The meaning of the sentences is unclear.
- Line 216: Please write C5aR instead of C5a receptor
- How can the increase in C5aR and CD59 gene expression in the early phase but decrease in the late phase be explained?
- The description of the results should be attenuated if they were not significantly different.
- The discussion would benefit from some sentences at the end how the knowledge gained from the study contributes to the understanding of tendon degeneration/healing/tendinopathy or what this offers for future investigations, treatment options,….
- A limitations part should be included, describing that e.g. high n-number variations could have avoided to detect significant differences on protein level whereas on gene expression level significances occurred. Additionally, investigations are based on tenocytes that come from healthy hamstring tendons. Would you expect a different result if the cells would be derived from tendinopathies/tendon rupture having an "inflammatory background"?
- General comment: Literature references and figure references in the text should be placed at the end of a sentence and not in between to allow a fluent reading (Example: lines 247, 249, ..).
Material and Methods:
- More information on the tenocyte donors should be included (age range, sex,…)
- Please give the rational of the used concentrations of C5a
- 4.4. The ELISA analysis is described very detailed with every washing step whereas the RNA isolation and cDNA synthesis refers the manual instructions. Please unify this. I would recommend not detailing the ELISA assay, as this is a standard methods that can be referred to the instruction manual and only individual details such as sample dilutions, standard range and detection systems should be mentioned.
- Line 340: Which other housekeeping genes were tested?
- Please give details of the TaqMan 340 Gene Expression Master Mix solution. Company?
- What is the corrected total cell fluorescence (CTCF)?
- Table 2: The heading of the last column should be "Used dilution" or vice versa the concentrations should be given in µg/ml.
- Table 2: Why 2 different Phalloidin Dyes are found in the table but only one is mentioned in the text?
- Statistics: It is described that a test for normal distribution was performed. How was the result? It is hard to believe that a normal distribution can be reached with n=3 or 4.
Conclusion:
- The conclusion would benefit from a final sentence highlighting the significance of the study.

Author Response
The authors would like to thank the reviewer for carefully reading the text and their constructive valuable comments. We modified the manuscript according to their suggestions and comments. A list of changes is reported below. All changes performed are indicated in red and underlined in the revised version of the manuscript. The manuscript has been carefully proof read. Please refer to our point by point reply below. We hope you will find the manuscript suitable for publication in IJMS now.
Sincerely,
Univ.-Prof. Dr. Gundula Schulze-Tanzil
(corresponding author)
Reviewer 2
The manuscript by Silawal and colleagues deals with an interesting and relevant topic in the field of tendon research. The results and conclusions drawn are partially based on non-significant data, which should be clearly attenuated overall in the text.
Response: The over-interpretation of non significant data has been eliminated.
I believe that a higher n-value for example in the protein data would lead to similar significant differences comparable to the gene expression data and I would recommend to repeat experiments to strengthen the conclusions.
Response: The reviewer´s comment is genuine. However, creating higher n-value would not be possible in the short time span provided for revision of only 7 days. If the revision of the manuscript is still not acceptable, the authors would obviously comply to the suggestion. However, this process would need several months in processing.
Parts of the text are unclear and I would recommend a professional proofreading.
Response: A proofreading of this manuscript by a native speaker has been done.
In the text a further publication by Silawal et al. is mentioned which is currently in press and is dealing with the same topic. Therefore, I would like to know the difference to the submitted manuscript.
Response: The other publication is completely different: An indirect co-culture of tenocytes with leukocytes subpopulations (peripheral blood mononuclear cells and neutrophils) with or without TNFα stimulation has been studied in an in vitro study (Silawal et al. 2021, in press) to understand the complement response in tenocytes. In this study we found an induction of TNFα and regulation of complement by leukocytes. The content of the other manuscript has been introduced in Line 54-58, page 2 in more detail.
Further detailed comments are found below.
Decision: major revision
Abstract:
The abstract is clearly written and highlights the most important results of the study.
- Line 24: I consider revising the sentence “Similarly, CD46 and …” as the word order is rather confusing.
- Response: The authors apologize for the confusion. The sentence was somehow distorted during the revision process. The abstract has been modified now.
- Line 28: It says that “C5a, released from damaged tendons”. Is this a result of the study, a hypothesis or is it known from literature? If it is known from literature, it is an important fact that should be mentioned at the beginning of the abstract as well as in the introduction.
- Response: The final sentence has been removed, since the complement activation with C5a is more a hypothesis. Further researches has to be performed in this field.
Introduction:
- The introduction could be in some parts tightened up, as there are doublings in the text that could be shortened (e.g. line 36-38 vs. line 45-48).
- Response: The introduction section has been revised and specifically, the redundant text has been eliminated.
Results:
- 1.
- Figure 2A: The schematic figure could be improved by drawing not only one cell per well, but some more.
- Response: More cells have been added into the wells.
- Figure 2B: the scale bar is missing in the control and 25ng/ml image.
- Response: The scale bar is implied to all the images in the image set.
- Figure 2C: The very small drop in cell vitality with highest standard deviation is described as “slight decrease”. I assume this is a small biological variation rather than a biological effect. Therefore, this should not be mentioned.
Response: Yes, the non-significant value as mentioned by the reviewer will not been emphasized. Hence, the sentence has been replaced in lines 105-106.
- Was the vitality test also performed for the 0.5 h and 4h time point and what was the result? And if not, why?
Response: Unfortunately, the earlier time points of investigation were not considered in the experimental setting for the vitality testing, but only the longest time point estimating that the end of the whole observation time would reflect the largest difference in vitality. We agree with the reviewer that a transient drop in vitality could also occur earlier.
2.2:
- The order of the text should be revised, so that at first figure 3A and B are described and later on C and D.
- Response: As suggested by the reviewer, the order of explaining the results of the respective graphs is switched from lines 148 -154 to lines 133 -138. During this process, the paragraph in lines 139-141 was also recognized as being rather a part of discussion. Hence, it was removed.
- Line 134: Figure 2 must be figure 3 and all following figures until figure 6 have the wrong numbering.
- Response: Thank you very much for indicating this mistake. The figure numbers have been corrected in the right order now.
- Figure 3: Description of the IL6 gene expression results are missing in the text.
- Response: It is mentioned now in lines 148-149 that IL-6 gene expression was not significantly affected by the C5a treatment.
- Regarding all gene expression graphs the y-axis should be displayed in a logarithmic scale to give down-regulations the same distance from 1 as up-regulations.
Response: The gene expression data (Figs. 3 and 4) is shown in logarithmic scale now.
- 3.
- Line 155-156: Mentioning the average protein values above all groups is confusing. What is this good for?
- Response: Except for the control the average protein values have been removed. Since the data in the diagram summarizing the TNFα protein concentrations are normalized we still mentioned the mean TNFα protein in the control group in relation to the total protein concentration to get an impression of the amount of TNFα protein.
- Describing changes on protein level in % is meaningless if the y-axis is given as fold changes. Changes should be described either as x-fold increase or the y-axis changed to %.
- Response: As suggested by the reviewer, the fold changes version has been maintained for the description part.
- No significant differences occurred on protein level (probably due to the low number of repetitions n=3) but the description is very detailed. The changes should not be over-interpreted and it should be mentioned that the result are not statistically significant. Compared to e.g. the TNFa gene expression were it is only described that “no further significant effect was observed” every little regulation is described on protein level. Why? If the authors believe that this has a significant biological relevance, the protein experiments should be repeated to increase the n-value and with this the power of the results.
- Response: The overstated interpretation has been completely removed.
- 4.:
- Again, the authors should avoid over-interpreting their data without significance. Please attenuate the statements.
- Response: The overstated interpretation of non-significant data has been removed.
- Discussion:
- The discussion would profit from a small paragraph at the beginning introducing again the aim and hypothesis of the study.
- Response: Lines 209-212 has been added to follow this advice of the reviewer.
- In other cells C5a induced apoptosis. Why was his not reached in your study? Was the concentration different? How were the concentrations defined? Could higher concentrations have led to a reduced cell vitality?
- Response: This may be due to the fact that the human tenocytes as connective tissue cells derived from a musculoskeletal tissue with low blood supply and hence, low presence of immune cells under normal conditions are more robust than the animal derived adrenomedullary or endothelial cells.
The question cannot be answered if higher concentrations would reduce vitality, since the highest concentration tested in this study was 100 ng/ml. Plasma concentration of 8.34 ± 2.05 ng/ml C5a have been reported (see introduction line 78). Under the conditions of the inflammatory phase of tissue injury in the presence of leukocytes the concentrations might be elevated. However, the tenocytes used in this study derived from healthy tendons of young donors, the results with tenocytes from impaired or tendinopathic tendons might differ displaying possibly a more sensitive response. This could present a further limitation of the study. To explain this in more detail lines 221-223 have been newly introduced.
- Line 215-219: The meaning of the sentences is
- Response: The lines were superfluous hence, have been removed, since they describe rather a scenario of a sepsis than of tendinopathy. The approach was not correct.
- Line 216: Please write C5aR instead of C5a receptor
Response: The whole sentence has been removed.
- How can the increase in C5aR and CD59 gene expression in the early phase but decrease in the late phase be explained?
- Response: The increase was only significant for CD59 and represented only a trend for C5aR. This displays an early response of tenocytes to synthesize C5aR and CD59 which probably recedes after time (4 h) as a negative feedback loop. This conclusion has been added to the discussion section (lines 231 and 244).
-
- The description of the results should be attenuated if they were not significantly different.
- Response: The overstated interpretation of non-significant data has been attenuated
- The discussion would benefit from some sentences at the end how the knowledge gained from the study contributes to the understanding of tendon degeneration/healing/tendinopathy or what this offers for future investigations, treatment options,….
- Response: Concluding sentences have been added in lines 278-283. See also the conclusion section on page 14, lines 420-423.
- A limitations part should be included, describing that e.g. high n-number variations could have avoided to detect significant differences on protein level whereas on gene expression level significances occurred. Additionally, investigations are based on tenocytes that come from healthy hamstring tendons. Would you expect a different result if the cells would be derived from tendinopathies/tendon rupture having an “inflammatory background”?
- Response: The limitations of this study – including the variation of n-numbers - have been summarized at the end of the discussion (lines 272-283). The aspect of a possible different behavior of tendinopathic tenocytes has also been introduced.
- General comment: Literature references and figure references in the text should be placed at the end of a sentence and not in between to allow a fluent reading (Example: lines 247, 249, ..).
- Response:
- The changes have been made according to reviewer´s recommendation
Material and Methods:
- More information on the tenocyte donors should be included (age range, sex)
- Response: The donors’ information has been added now.
- Please give the rational of the used concentrations of C5a
- Response: The concentrations were selected based on the plasma concentration of 8.3 ng reported in healthy individuals (Lechner J et al., 2016) and the assumption of an amplification of it during inflammation and under tendinopathic conditions. It is within the range of other studies cited (Tsai et al., 2019). It is described in lines 219-221 now.
- 4. The ELISA analysis is described very detailed with every washing step whereas the RNA isolation and cDNA synthesis refers the manual instructions. Please unify this. I would recommend not detailing the ELISA assay, as this is a standard methods that can be referred to the instruction manual and only individual details such as sample dilutions, standard range and detection systems should be mentioned.
Response: Yes, the standard steps have been shorted as the steps are performed referred to manufacturer´s instructions. Line 341-342.
- Line 340: Which other housekeeping genes were tested?
Response: HPRT served as the most stable reference gene tested before compared to β-actin, TATA-binding protein or Glyceraldehyde 3-phosphate dehydrogenase. This comparison of reference genes before performing PCR analysis with HPRT has been introduced in Line 329 – 330.
- Please give details of the TaqMan 340 Gene Expression Master Mix solution. Company?
- Response: The product detail has been added. Line 331.
- What is the corrected total cell fluorescence (CTCF)?
- Response: CTCF = Integrated Density – (Area of selected cell X Mean fluorescence of background readings). This has been introduced in the line 366.
- Table 2: The heading of the last column should be “Used dilution” or vice versa the concentrations should be given in µg/ml.
- Response: The heading has been changed into “used dilution”
- Table 2: Why 2 different Phalloidin Dyes are found in the table but only one is mentioned in the text?
- Response: That´s true! The redundant Phalloidin 488 dye has been removed.
- Statistics: It is described that a test for normal distribution was performed. How was the result? It is hard to believe that a normal distribution can be reached with n=3 or 4.
- Response: Shapiro Wilk normality test was performed. The reviewer´s remark is absolutely correct. C5aR protein expression (4 h/100 ng/ml) and TNFα protein expression (24 h) values were not in normal distribution. This results are hence not qualified for the interpretation. Line
Conclusion:
- The conclusion would benefit from a final sentence highlighting the significance of the study.
Response: we revised the conclusion. “This study was performed in a non-inflammatory pre-condition and void of external stressors therefore, helps to understand the complement as well as TNFα related response to anaphylatoxin on tenocytes.” Lines 422-425.

Round 2
Reviewer 2 Report
The authors nicely addressed most of the questions and the content of the manuscript improved. The recommendation of repetition of further experiments to increase the n-value and support the conclusions drawn in the manuscript was rejected by the authors due to the limited time of 7 days for the review, which avoided further experiments. As the first reviewer did not complain about the low n-value of some experiments, I would leave it to the editor’s decision whether the experimental significance or the timely compliance with the review deadline is more important. Next to this point, I have some minor specific comments to improve the manuscript.
Figure 3:
- The stars indicating the significance in C5aR gene expression of the 24h time points seem to be placed wrong, as it refers to the 100ng/ml concentration in the text.
Figure 3 and 4:
- In the legends the control group is grey, but the control is now given by the reference line at 1. This should be corrected in the figures (delete grey box) and figure legends.
CTCF:
- I still don’t understand the calculation of the CTCF. Why the area of the cell was multiplied by the mean background fluorescence? Shouldn’t it be the mean fluorescence of the cell
Author Response
Dear Editor, 13th March 2021
The authors would like to thank the reviewer again for carefully reading the manuscript and very valuable comments. We modified the manuscript according to the reviewer suggestions with a list of changes shown below. All changes performed are indicated in red and underlined in the revised version of the manuscript. We hope you will find this manuscript suitable for publication in “Int J Mol Sci”. Please do not hesitate to contact me anytime for questions regarding this manuscript.
Sincerely,
Univ.-Prof. Dr. Gundula Schulze-Tanzil
(corresponding author)
Reviewer 2
The authors nicely addressed most of the questions and the content of the manuscript improved. The recommendation of repetition of further experiments to increase the n-value and support the conclusions drawn in the manuscript was rejected by the authors due to the limited time of 7 days for the review, which avoided further experiments. As the first reviewer did not complain about the low n-value of some experiments, I would leave it to the editor’s decision whether the experimental significance or the timely compliance with the review deadline is more important. Next to this point, I have some minor specific comments to improve the manuscript.
Figure 3:
- The stars indicating the significance in C5aR gene expression of the 24h time points seem to be placed wrong, as it refers to the 100ng/ml concentration in the text.
- The authors thank the reviewer once again for the thorough review. Indeed the changes of the diagrams into a new display formate led to the mistakes. Thankfully, the stars indicating the significance in C5aR gene expression as well as in that of CD55 have been positioned in the correct order now.
Figure 3 and 4:
- In the legends the control group is grey, but the control is now given by the reference line at 1. This should be corrected in the figures (delete grey box) and figure legends.
- Also the legends have been modified in Figure 3 & 4.
CTCF:
- I still don’t understand the calculation of the CTCF. Why the area of the cell was multiplied by the mean background fluorescence? Shouldn’t it be the mean fluorescence of the cell
- Since the measured intensity (integrated density; also measured in respect to area of the cells) is rather an overlap of the actual intensity (CTCF) and the background fluorescence. After deriving the mean fluoresence background readings, this value has to be eliminated from the measured mean fluorescence of the cells. CTCF = Integrated Density – (Area of selected cell X Mean fluorescence of background readings).
https://www.researchgate.net/publication/309350959_Calculate_the_Corrected_Total_Cell_Fluorescence_CTCF
- The lines 390-398 have been modified accordingly and literature references suppoting the described procedure have been added now.
